# Metagenomics workflow for hybrid assembly, differential coverage binning, metatranscriptomics and pathway analysis (MUFFIN)

**Renaud Van Damme** [1,2]*, **Martin Hölzer** [3], **Adrian Viehweger** [3,4], **Bettina Müller** [1], **Erik Bongcam-Rudloff** [2], **Christian Brandt** [2,5]

**1** Department of Molecular Sciences, Swedish University of Agricultural Sciences, Uppsala, Sweden, **2** Department Animal Breeding and Genetics, Bioinformatics section, Swedish University of Agricultural Sciences, Uppsala, Sweden, **3** RNA Bioinformatics and High-Throughput Analysis, Friedrich Schiller University Jena, Jena, Germany, **4** Department of Medical Microbiology, University Hospital Leipzig, Leipzig Germany, **5** Institute for Infectious Diseases and Infection Control, Jena University Hospital, Jena, Germany

* renaud.van.damme@slu.se

**Data Availability Statement:** All subset files for testing the pipeline are available from https://osf.io/m5czv/ MUFFIN is available at https://github.com/

## Abstract

Metagenomics has redefined many areas of microbiology. However, metagenome-assembled genomes (MAGs) are often fragmented, primarily when sequencing was performed with short reads. Recent long-read sequencing technologies promise to improve genome reconstruction. However, the integration of two different sequencing modalities makes downstream analyses complex. We, therefore, developed MUFFIN, a complete metagenomic workflow that uses short and long reads to produce high-quality bins and their annotations. The workflow is written by using Nextflow, a workflow orchestration software, to achieve high reproducibility and fast and straightforward use. This workflow also produces the taxonomic classification and KEGG pathways of the bins and can be further used for quantification and annotation by providing RNA-Seq data (optionally). We tested the workflow using twenty biogas reactor samples and assessed the capacity of MUFFIN to process and output relevant files needed to analyze the microbial community and their function. MUFFIN produces functional pathway predictions and, if provided *de novo* metatranscript annotations across the metagenomic sample and for each bin. MUFFIN is available on github under GNUv3 licence: https://github.com/RVanDamme/MUFFIN.

## Author summary

Determining the entire DNA of environmental samples (sequencing) is a fundamental approach to gain deep insights into complex bacterial communities and their functions. However, this approach produces enormous amounts of data, which makes analysis time intense and complicated. We developed the Software "MUFFIN," which effortlessly untangle the complex sequencing data to reconstruct individual bacterial species and determine their functions. Our software is performing multiple complicated steps in

RVanDamme/MUFFIN under GNU General Public License version 3.

**Funding:** This study was funded by the Deutsche Forschungsgemeinschaft (DFG, German Research Foundation) – BR 5692/1-1 and BR 5692/1-2. This material is based upon work supported by Google Cloud. BM was funded by FORMAS, grant number 942-2015-1008. MH is supported by the Collaborative Research Centre AquaDiva (CRC 1076 AquaDiva) of the Friedrich Schiller University Jena, funded by the DFG. MH appreciates the support of the Joachim Herz Foundation by the add-on fellowship for interdisciplinary life science. The funders had no role in study design, data collection and analysis, decision to publish, or preparation of the manuscript.

**Competing interests:** The authors have declared that no competing interests exist.

parallel, automatically allowing everyone with only basic informatics skills to analyze complex microbial communities.

For this, we combine two sequencing technologies: "long-sequences" (nanopore, better reconstruction) and "short-sequences" (Illumina, higher accuracy). After the reconstruction, we group the fragments that belong together ("binning") via multiple approaches and refinement steps while also utilizing the information from other bacterial communities ("differential binning"). This process creates hundreds of "bins" whereas each represents a different bacterial species with a unique function. We automatically determine their species, assess each genome's completeness, and attribute their biological functions and activity ("transcriptomics and pathways"). Our Software is entirely freely available to everyone and runs on a good computer, compute cluster, or via cloud.

This is a *PLOS Computational Biology* Software paper.

## Introduction

Metagenomics is widely used to analyze the composition, structure, and dynamics of microbial communities, as it provides deep insights into uncultivatable organisms and their relationship to each other [1–5]. In this context, whole metagenome sequencing is mainly performed using short-read sequencing technologies, predominantly provided by Illumina. Not surprisingly, the vast majority of tools and workflows for the analysis of metagenomic samples are designed around short reads. However, long-read sequencing technologies, as provided by PacBio or Oxford Nanopore Technologies (ONT), retrieve genomes from metagenomic datasets with higher completeness and less contamination [6]. The long-read information bridges gaps in a short-read-only assembly that often occur due to intra- and interspecies repeats [6]. Complete viral genomes can be already identified from environmental samples without any assembly step via nanopore-based sequencing [7]. Combined with a reduction in cost per gigabase [8] and an increase in data output, the technologies for sequencing long reads quickly became suitable for metagenomic analysis [9–12]. In particular, with the MinION, ONT offers mobile and cost-effective sequencing device for long reads that paves the way for the real-time analysis of metagenomic samples. Currently, the combination of both worlds (long reads and high-precision short reads) allows the reconstruction of more complete and more accurate metagenome-assembled genomes (MAGs) [6].

One of the main challenges and bottlenecks of current metagenome sequencing studies is the orchestration of various computational tools into stable and reproducible workflows to analyze the data. A recent study from 2019 involving 24,490 bioinformatics software resources showed that 26% of all these resources are not currently online accessible [13]. Among 99 randomly selected tools, 49% were deemed 'difficult to install,' and 28% ultimately failed the installation procedure. For a large-scale metagenomics study, various tools are needed to analyze the data comprehensively. Thus, already during the installation procedure, various issues arise related to missing system libraries, conflicting dependencies and environments, or operating system incompatibilities. Even more complicating, metagenomic workflows are computing intense and need to be compatible with high-performance compute clusters (HPCs), and thus different workload managers such as SLURM or LSF. We combined the workflow manager Nextflow [14] with virtualization software (so-called 'containers') to generate reproducible results in various working environments and allow full parallelization of the workload to a higher degree.

Several workflows for metagenomic analyses have been published, including MetaWRAP (v1.2.1) [15], Anvi'o [16], SAMSA2 [17], Humann [18], MG-Rast [19], ATLAS [20], or Sunbeam [21]. Unlike those, MUFFIN allows for a hybrid metagenomic approach combining the strengths of short and long reads. It ensures reproducibility through the use of a workflow manager and reliance on either install-recipes (Conda [22]) or containers (Docker [23], Singularity).

## Design and implementation

MUFFIN integrates state-of-the-art bioinformatic tools via Conda recipes or Docker/Singularity containers for the processing of metagenomic sequences in a Nextflow workflow environment (Fig 1). MUFFIN executes three steps subsequently or separately if intermediate results, such as MAGs, are available. As a result, a more flexible workflow execution is possible. The three steps represent common metagenomic analysis tasks and are summarized in Fig 1:

1. Assemble: Hybrid assembly and binning

2. Classify: Bin quality control and taxonomic assessment

3. Annotate: Bin annotation and KEGG pathway summary

The workflow takes paired-end Illumina reads (short reads) and nanopore-based reads (long reads) as input for the assembly and binning and allows for additional user-provided read sets for differential coverage binning. Differential coverage binning facilitates genome bins with higher completeness than other currently used methods [24]. Step 2 will be executed automatically after the assembly and binning procedure or can be executed independently by providing MUFFIN a directory containing MAGs in FASTA format. In step 3, paired-end RNA-Seq data can be optionally supplemented to improve the annotation of bins.

On completion, MUFFIN provides various outputs such as the MAGs, KEGG pathways, and bin quality/annotations. Additionally, all mandatory databases are automatically downloaded and stored in the working directory or can be alternatively provided via an input flag.

**Step 1—Assemble: Hybrid assembly and binning.** The first step (**Assembly and binning**) uses metagenomic nanopore-based long reads and Illumina paired-end short reads to obtain high-quality and highly complete bins. The short-read quality control is operated using fastp (v0.20.0) [25]. Optionally, Filtlong (v0.2.0) [26] can be used to discard long reads below a length of 1000 bp. The hybrid assembly can be performed according to two principles, which differ substantially in the read set to begin with. The default approach starts from a short-read assembly where contigs are bridged via the long reads using metaSPAdes (v3.13.2) [27–29]. Alternatively, MUFFIN can be executed starting from a long-read-only assembly using metaFlye (v2.8) [30,31] followed by polishing the assembly with the long reads using Racon (v1.4.13) [32] and medaka (v1.0.3) [33] and finalizing the error correction by incorporating the short reads using multiple rounds of Pilon (v1.23) [34]. Both approaches should be chosen based on the available amount of raw read data available to users. E.g., if more short read data is available, meta-spades should be the choice (long reads are "supplemental"). If more long-read data is available, e.g.,> 15 Gigabases (corresponds to a full MinION or GridION flow cell) [35] flye should be used as the assembly approach.

Binning is one of the most crucial steps during metagenomic analysis besides assembly. Therefore, MUFFIN combines three different binning software tools, respectively CONCOCT (v1.1.0) [36], MaxBin2 (v2.2.7) [37], and MetaBAT2 (v2.13) [38] and refine the obtained bins via MetaWRAP (v1.3) [15]. The user can provide additional read data sets (short or long

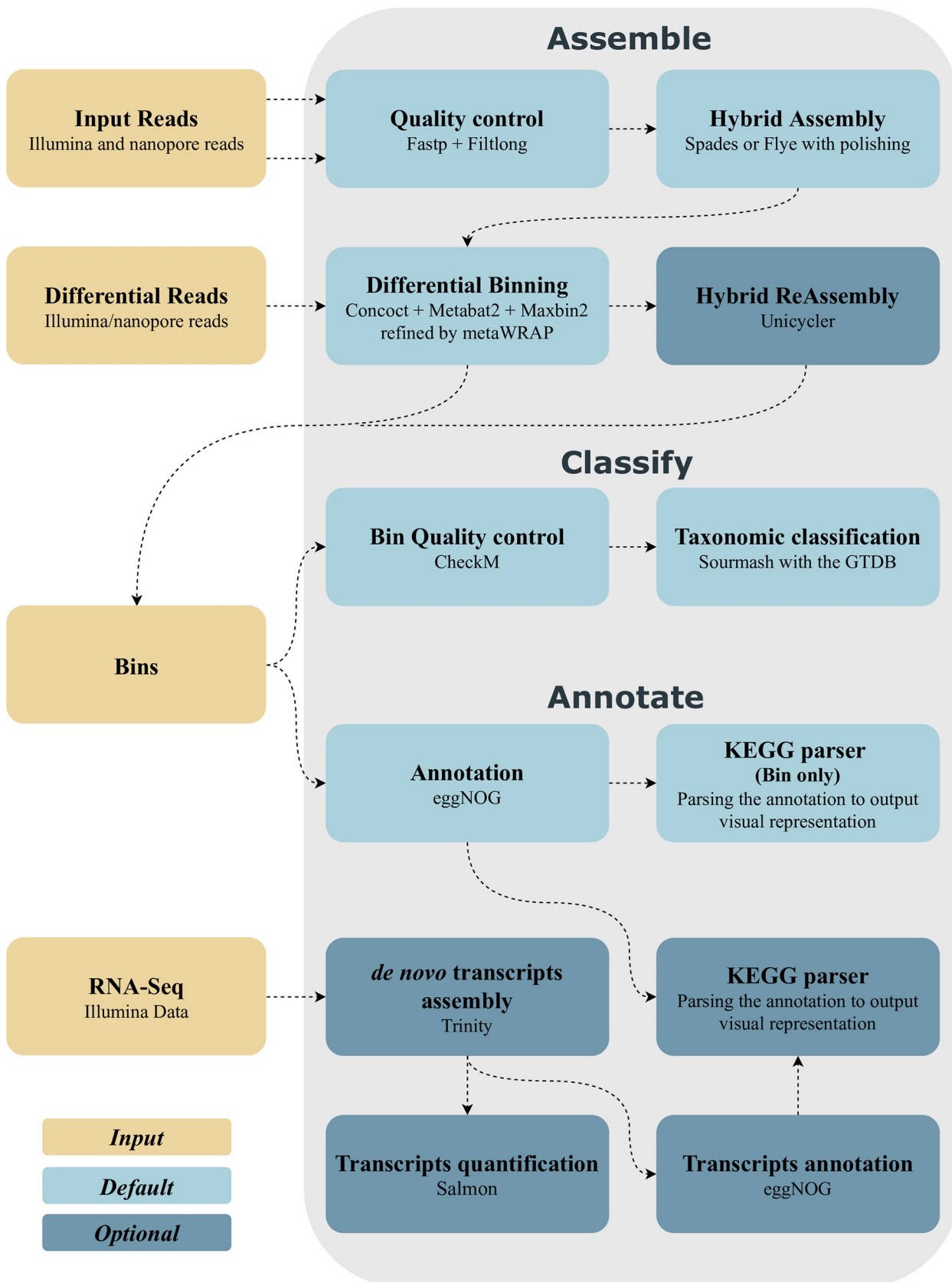

**Fig 1. Simplified overview of the MUFFIN workflow.** All three steps (Assemble, Classify, Annotate) from top to bottom are shown. The RNA-Seq data for Step 3 (Annotate) is optional. Differential reads are other read data sets that are solely used for "differential coverage binning" to improve the overall binning performance.

reads) to perform automatically differential coverage binning to assign contigs to their bins better.

Moreover, an additional reassembly of bins has shown the capacity to increase the completeness and N50 while decreasing the contamination of some bins [15]. Therefore, MUFFIN allows for an optional reassembly to improve the continuity of the MAGs further. This reassembly is performed by retrieving the reads belonging to one bin and doing an assembly with Unicycler (v0.4.7) [39]. As each reassembly might improve or worsen each bin, this process is optional and therefore deactivated by default. Individual manual curation is necessary by the user to compare each bin before and after reassembly, as described by Uritskiy *et al.* [15].

To support a transparent and reproducible metagenomics workflow, all reads that cannot be mapped back to the existing high-quality bins (after the refinement) are available as an output for further analysis. These "unused" reads could be further analyzed by other tools such as Kraken2 [40], Kaiju [41], or centrifuge [42] for read classification, "What the Phage" [43] to search for phages, mi-faser [44] for functional annotation of the reads or even use these reads as a new input to run MUFFIN.

**Step 2—Classify: Bin quality control and taxonomic assessment.** In the second step (**Bin quality control and taxonomic assessment**), the quality of the bins is evaluated with CheckM (v1.1.3) [45] followed by assigning a taxonomic classification to the bins using sourmash (v2.0.1) [46] and the Genome Taxonomy Database (GTDB release r89) [47]. The GTDB was chosen as it contains many unculturable bacteria and archaea–this allows for monophyletic species assignments, which other databases do not assure [35,48]. Moreover, the coherent taxonomic classifications and more accurate taxonomic boundaries (e.g., for class, genus, etc.) proposed by GTDB substantially increases the general classification accuracy [48]. The user can also analyze other bin sets in this step regardless of their origin by providing a directory with multiple FASTA files (bins).

**Step 3—Annotate: Bin annotation and KEGG pathway summary.** The last step of MUFFIN (**Bin annotation and output summary**) comprises the annotation of the bins using eggNOG-mapper (v2.0.1) [49] and the eggNOG database (v5) [50]. If RNA-Seq data of the metagenome sample is provided (Illumina, paired-end), quality control using fastp (v0.20.0) [25] and a *de novo* metatranscript assembly using Trinity (v2.9.1) [51] followed by quantification of the metatranscripts by mapping of the RNA-seq reads using Salmon (v1.0) [52] are performed. Lastly, the metatranscripts are annotated using eggNOG-mapper (v2.0.1) [49]. Again, the annotation by eggnog-mapper provides a wide array of annotation information such as the GO terms, the NOG terms, the BiGG reaction, CAZy, KEGG orthology, and pathways.

These gene annotations are parsed and visualized in KEGG pathways for each sample and bin. The expression of low and high abundant genes present in the bins is shown. If only bin sets are provided without any RNA-Seq data, the pathways of all the bins are created based on gene presence alone. The KEGG pathway results are summarized in detail as interactive HTML files (example snippet: Fig 2).

Like step 2, this step can be directly performed with a bin set created via another workflow.

## Running MUFFIN and version control

MUFFIN (V1.0.3, 10.5281/zenodo.4296623) requires only two dependencies, which allows an easy and user-friendly workflow execution. One of them is the workflow management system

| Sample overview — Summary of the pathways and orthologs on a sample level | Pathway highlight 1 | ... | Pathway highlight 5 | Bins Compositions |
|---|---|---|---|---|
| | Pathway name + link to the highlight 1 | ... | Pathway name + link to the highlight 5 | Bin 1 [X orthologs identical to RNAseq, Y orthologs not found in RNAseq]; Bin 2 [ X , Y ]; ...... Bin n° [ X , Y ] |

| Bin overview — Summary of the pathways and orthologs for each bin | Pathway highlight 1 to 4 | List of orthologs present in both RNAseq and the bin | List of orthologs only present in the bin |
|---|---|---|---|
| | Pathway name + link to the highlight | Ortholog name +link Ortholog name +link ...... Ortholog name +link | Ortholog name +link Ortholog name +link .... Ortholog name +link |

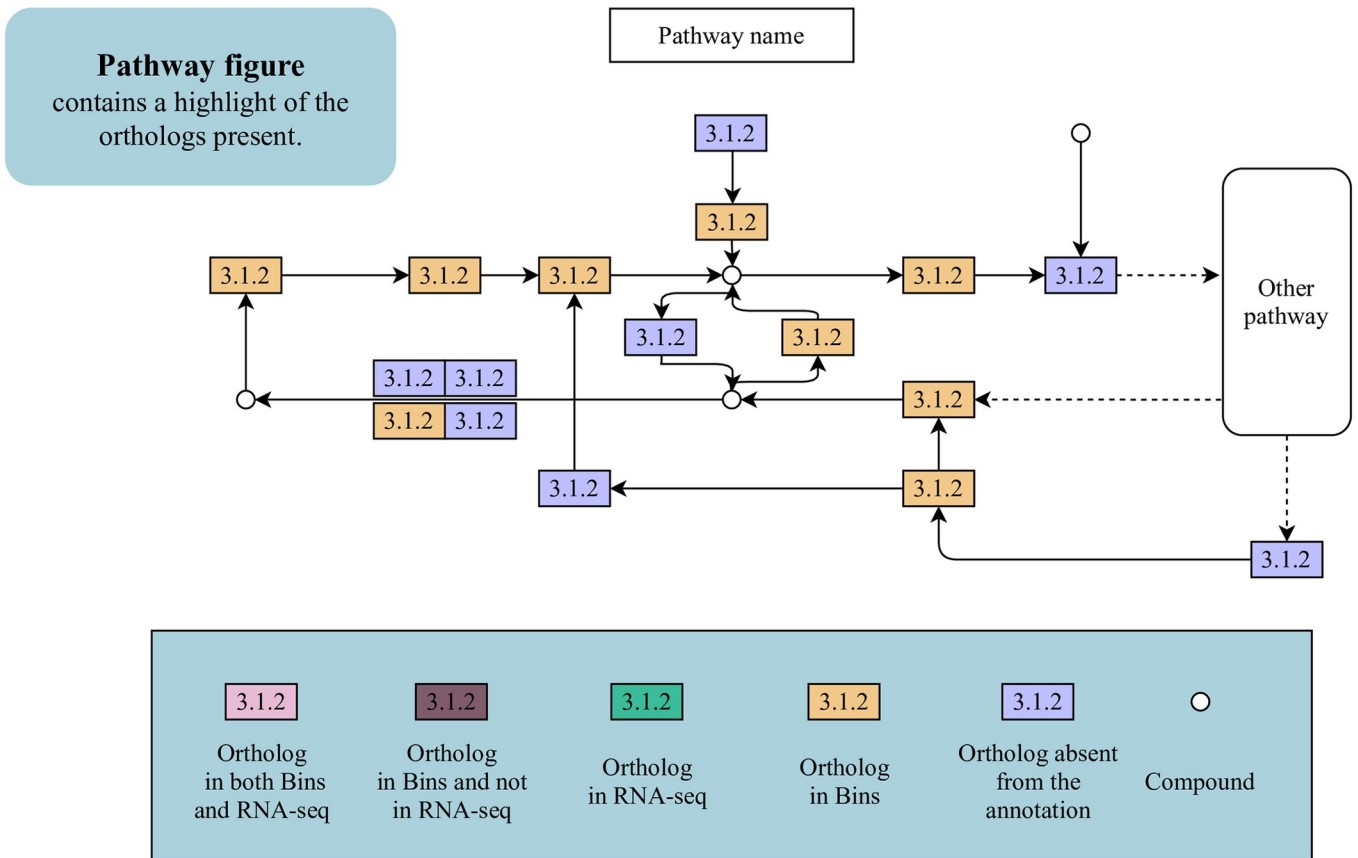

**Pathway figure** contains a highlight of the orthologs present.

Fig 2. Example snippets of the sub-workflow results of step 3 (Annotate).

Nextflow [14] (version 20.07+), and the other can be either Conda [20][22] as a package manager or Docker [23] / Singularity to use containerized tools. A detailed installation process is available on https://github.com/RVanDamme/MUFFIN. Each MUFFIN release specifies the Nextflow version it was tested on, but any version of MUFFIN V1.0.2+ will work with nextflow version 20.07+. A Nextflow-specific version can always be directly downloaded as an executable file from https://github.com/nextflow-io/nextflow/releases, which can then be paired with a compatible MUFFIN version via the -r flag.

## Results

We chose Nextflow for the development of our metagenomic workflow because of its direct cloud computing support (Amazon AWS, Google Life Science, Kubernetes), various ready-to-use batch schedulers (SGE, SLURM, LSF), state-of-the-art container support (Docker, Singularity), and accessibility of a widely used software package manager (Conda). Moreover, Nextflow [14] provides a practical and straightforward intermediary file handling with process-specific work directories and the possibility to resume failed executions where the work ceased. Additionally, the workflow code itself is separated from the 'profile' code (which contains Docker, Conda, or cluster related code), which allows for a convenient and fast workflow adaptation to different computing clusters without touching or changing the actual workflow code.

The entire MUFFIN workflow was executed on 20 samples from the Bioproject PRJEB34573 (available at ENA or NCBI) using the Cloud Life Sciences API (google cloud) with docker containers. This metagenomic bioreactor study provides paired-end Illumina and nanopore-based data for each sample [35]. We used five different Illumina read sets of the same project for differential coverage binning, and the workflow runtime was less than two days for all samples. MUFFIN was able to retrieve 1122 MAGs with genome completeness of at least 70% and contamination of less than 10% (Fig 3). In total, MUFFIN retrieved 654 MAGs with genome completeness of over 90%, of which 456 have less than 2% contamination out of the 20 datasets. For comparison, a recent study was using 134 publicly available datasets from different biogas reactors and retrieved 1,635 metagenome-assembled genomes with genome completeness of over 50% [53].

Exemplarily, we investigated the impact of additional reassembly of each bin for five samples (Fig 3). The N50 was increased by an average of 6–7 fold across all samples. Twenty-six bins of the five samples had an N50 ranging between 1 to 3 Mbases. Reassembly of bins has shown the capacity to increase the completeness and N50 while decreasing the contamination of some bins [15]. This is in line with our samples as some bins benefit more from this step than others. In general, while we observed a general increase in N50 for most bins, the genome quality based on checkM metrics (completeness, contamination) was slightly increasing or decreasing for individual bins.

## Discussion

The analysis of metagenomic sequencing data evolved as an emerging and promising research field to retrieve, characterize, and analyze organisms that are difficult to cultivate. There are numerous tools available for individual metagenomics analysis tasks, but they are mainly developed independently and are often difficult to install and run. The MUFFIN workflow gathers the different steps of a metagenomics analysis in an easy-to-install, highly reproducible, and scalable workflow using Nextflow, which makes them easily accessible to researchers.

MUFFIN utilizes the advantages of both sequencing technologies. Short-reads provide a better representation of low abundant species due to their higher coverage based on read count. Long-reads are utilized to resolve repeats for better genome continuity. This aspect is

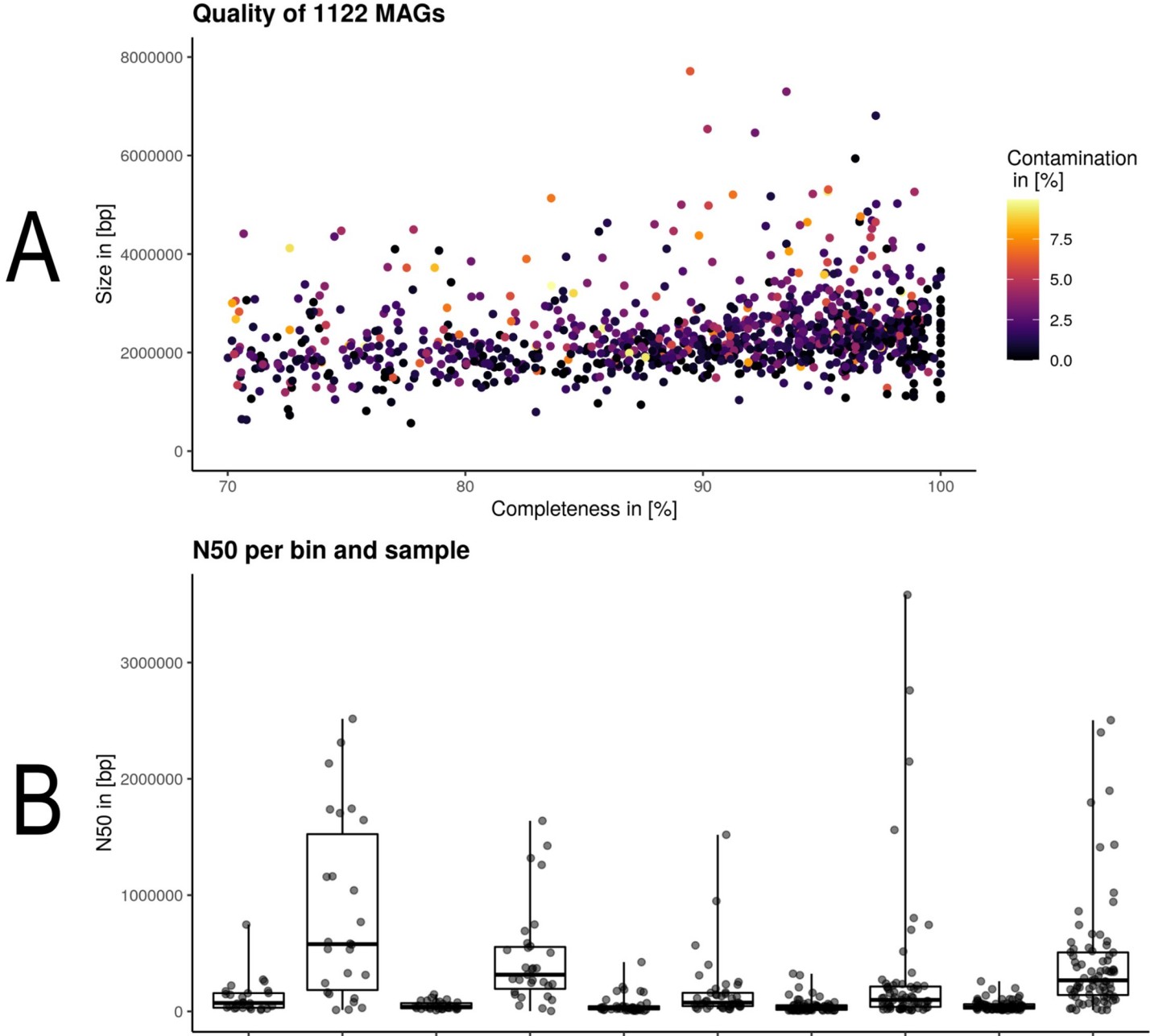

**Fig 3. Quality of meta-assembled genomes (MAGs).** [A] Quality overview of 1122 MAGs by plotting size to completeness and coloring based on contamination level. [B] N50 comparison between each bin of five selected samples from the Bioproject PRJEB34573 before and after individual bin reassembly.

further utilized via the final reassembly step after binning, which is an optional step due to the additional computational burden which solely aims to improve genome continuity.

Another critical aspect is the full support of differential binning, for both long and short reads, via a single input option. The additional coverage information from other read sets of similar habitats allows for the generation of more concise bins with higher completeness and less contamination because more coverage information is available for each binning tool to decide which bin each contig belongs to.

With supplied RNA-Seq data, MUFFIN is capable of enhancing the pathway results present in the metagenomic sample by incorporating this data as well as the general expression level of the genes. Such information is essential to further analyze metagenomic data sets in-depth, for example, to define the origin of a sample or to improve environmental parameters for production reactors such as biogas reactors. Knowing whether an organism expresses a gene is a crucial element in deciding whether more detailed analysis of that organism in the biotope where the sample was taken is necessary or not.

MUFFIN utilizes a large number of tools to provide a comprehensive analysis of metagenomics samples. The associated tools were mainly chosen based on benchmark performance, e.g., assembly [29,31,54–56], polishing [55], binning [15], annotation for pathways [49], taxonomic classification [47], however stability and workflow compatibility was also an important factor to consider. Due to the modular coding structure of nextflow DSL2 language, MUFFIN can quickly adapt towards better tools or improved versions if necessary, in the future.

MUFFIN executes a de novo assembly of the RNA-seq reads instead of a mapping of the reads against the MAGs to avoid bias and error during the mapping. Indeed, not all the DNA reads were assembled or binned and present in the last step (annotation). Thus we might miss transcripts on the sample level. In addition, for similar genes, it's impossible to know to which organism the reads should map to. By using metatranscripts and comparing the annotations of the metatranscripts to the annotation of the MAGs, we avoid those issues.

## Availability and future directions

MUFFIN is an ongoing workflow project that gets further improved and adjusted. The modular workflow setup of MUFFIN using Nextflow allows for fast adjustments as soon as future developments in hybrid metagenomics arise, including the pre-configuration for other workload managers. MUFFIN can directly benefit from the addition of new bioinformatics software such as for differential expression analysis and short-read assembly that can be easily plugged into the modular system of the workflow. Another improvement is the creation of an advanced user and wizard user configuration file, allowing experienced users to tweak the different parameters of the different software as desired.

MUFFIN will further benefit from different improvements, in particular by graphically comparing the generated MAGs via a phylogenetic tree. Furthermore, a convenient approach to include negative controls is under development to allow the reliable analysis of super-low abundant organisms in metagenomic samples.

MUFFIN is publicly available at https://github.com/RVanDamme/MUFFIN under the GNU general public license v3.0. Detailed information about the program versions used and additional information can be found in the GitHub repository. All tools used by MUFFIN are listed in the S1 Table. The Docker images used in MUFFIN are prebuilt and publicly available at https://hub.docker.com/u/nanozoo, and the GTDB formatted for sourmash (v2.0.1)[46] usage is publicly available at https://osf.io/m5czv/. The MAGs produced by the 20 samples; the template of the output of MUFFIN (README_output.txt); the subset data use in the test profile of MUFFIN (subset_data.tar.gz); and the results of MUFFIN on the subset data with and without RNA using both flye and spades are also available at https://osf.io/m5czv/. The Version of MUFFIN presented in this paper is (V1.0.3, 10.5281/zenodo.4296623).

## Supporting information

**S1 Table. List of the MUFFIN task, the softwares and versions.**
(XLSX)

## Acknowledgments

We want to thank Hadrien Gourlé and Moritz Buck for the valuable insights into metagenomic analysis and annotation.

## Author Contributions

**Conceptualization:** Renaud Van Damme, Christian Brandt.

**Data curation:** Renaud Van Damme, Bettina Müller.

**Formal analysis:** Renaud Van Damme, Christian Brandt.

**Funding acquisition:** Christian Brandt.

**Investigation:** Renaud Van Damme.

**Methodology:** Renaud Van Damme, Martin Hölzer, Adrian Viehweger, Christian Brandt.

**Project administration:** Christian Brandt.

**Resources:** Bettina Müller, Erik Bongcam-Rudloff.

**Software:** Renaud Van Damme, Martin Hölzer, Adrian Viehweger, Erik Bongcam-Rudloff, Christian Brandt.

**Supervision:** Christian Brandt.

**Validation:** Renaud Van Damme, Bettina Müller.

**Visualization:** Renaud Van Damme.

**Writing – original draft:** Renaud Van Damme.

**Writing – review & editing:** Martin Hölzer, Adrian Viehweger, Erik Bongcam-Rudloff, Christian Brandt.

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
