## [Decision Letter · Decision Letter 0]

26 Sep 2020

Dear %TITLE% Van Damme,

Thank you very much for submitting your manuscript "Metagenomics workflow for hybrid assembly, differential coverage binning, transcriptomics and pathway analysis (MUFFIN)" for consideration at PLOS Computational Biology.

As with all papers reviewed by the journal, your manuscript was reviewed by members of the editorial board and by several independent reviewers. In light of the reviews (below this email), we would like to invite the resubmission of a significantly-revised version that takes into account the reviewers' comments.

We cannot make any decision about publication until we have seen the revised manuscript and your response to the reviewers' comments. Your revised manuscript is also likely to be sent to reviewers for further evaluation.

Sincerely,

Mihaela Pertea

Software Editor

PLOS Computational Biology

Mihaela Pertea

Software Editor

PLOS Computational Biology

Reviewer's Responses to Questions

**Comments to the Authors:**

Reviewer #1: This paper describes MUFFIN, a workflow application for binning MAGs

from shotgun metagenome data generated from both long and short-read

sequencing platforms.

Overall, the paper is well written, reasonably thorough, and describes

a workflow that is certainly needed. We are not aware of competing

workflows that deal with long reads, and we feel that Muffin is poised

to become a widely used tool!

Despite the overall quality of the paper, we do think there are a number

of things that could be reworded, clarified, and revised. Please see

below.

Major comments --

While ONT sequencing is certainly a fast-advancing technology, in our

limted experience it seems still to be mostly suitable for relatively

low complexity communities, and this paper glosses over its current

limitations. Perhaps we are wrong? If so, more references would be

useful (than just the one). Regardless it would be good to qualify the

statement here to make it clear that e.g. reference 6 is about hybrid

assembly, and not purely about de novo long read sequencing in

isolation. And/or it might be good to provide some minimal guidance

(perhaps elsewhere in the paper) as to how many reads from which ONT

platform would be useful.

line 108, "Binning is the most crucial step..." We are not sure that

is true - certainly assembly would seem to be important, for example.

Perhaps "a crucial step"?

A useful addition would be some discussion of why the tools chosen were

chosen. Perhaps there was no reason other than that "they worked in our

hands" (which is perfectly fine) but if there is more to add, please do!

It might also be good to comment on whether the workflow is flexible

enough to "swap in" different tools.

Line 179, are there downsides to this re-assembly? Perhapos chimerism/loss

of microdiversity? It would also be good to discuss why additional bins

were able to be constructed on a second round, after being missed in the

first round.

Perhaps comment on why RNAseq reads are assembled instead of being mapped?

What is being mapped to whom, in any case, line 137? Are the de novo

metagenome assemblies/MAGs used at all here?

Eggnog outputs a lot of information other than just KEGG... perhaps

some mentions of these would be good? (GO terms, NOG terms, suggested

taxonomy, etc)

Minor or easily resolved comments --

We suggest citing ATLAS and Sunbeam, which are short-read-focused workflows

for MAG extraction.

Please make an archival copy of the version of Muffin used in this

paper and provide the DOI; this can easily be done by connecting Zenodo

to GitHub and making a new release on GitHub.

line 44, 'such' can be removed.

In Figure 1, what does the word "Differential" (in Differential Reads as an

input) mean?

line 110, 'refine' should be plural

line 120, maybe suggest some additional tools beyond running

Muffin again? mifaser, for example?

line 126, sourmash v3.5 was released recently - perhaps upgrade past an

alpha version?

line 182, "more _from_ this step"

line 198, "whereas" should be rephrased - maybe refer to both

technologies? Something like "MUFFIN takes advantage of short reads

for abundance estimation and long reads for their ability to resolve

repeats" or something .

line 222, suggest removing both "all" words.

line 237, "Annotation" should not be capitalized.

Figure 3A colors make it hard to see the most contaminated bins. Maybe

invert the color scale? or choose the inferno palate instead of

viridis.

We suggest reporting not only the MAGs/length etc that MUFFIN

recovered, but also how many reads weren't assembled/analyzed/didn't make it

into the MAGs.

Reviewer #2: Long reads are more and more common for metagenomics. MUFFIN is a pipeline integrating short reads, long reads and RNA seq reads for functional annotation of metagenomes. It aggregates the results into Kegg pathways and produces nice html reports. It is great that the authors provided multiple executer systems. The installation instruction is clear, and the executor can be specified easily.

Major concerns:

1. Unable to run MUFFIN

I tried to run muffin on a Linux cluster with different combinations of executors and backends without success. The executed commands together with the error logs are attached.

2. More documentation to customize the cluster execution.

It is impossible to make a tool that can be run on any cluster system. The authors did a good job by providing solutions for slurm and google could. However, if they claim that Muffin can be executed on other systems (L160) they should provide profiles or document how users can create execution profiles for their clusters. It would also be helpful to explain how to configure the profiles.

3. Multiple samples.

It is not completely clear how a user can run Muffin with multiple samples. Does the user need to run muffin on each sample separately and define the optional other samples for differential binning? Is there no easier way to run Muffin on a set of samples? How can a user compare different samples? The main output of Muffins is the quantification of KEGG pathways based on MAGs (and RNA seq data) but the MAGs are sample-specific and not directly comparable.

4. RNA-seq:

If I understand correctly the RNAseq data comes from a microbial community, not from a single genome. In this case, the term “metatranscriptome” would be more appropriate.

Trinity was originally not designed for meta-transcriptomes but works relatively good (10.1186/2049-2618-2-39). Could you discuss the challenges of assembly for metatranscriptomes and why you used genome-independent transcriptome assembly if you have MAGs available?

5. Pathway aggregation.

Muffin provides nice html reports with of the KEGG pathways and the genes present in the MAGs and or the RNA seq data. Unfortunately, the gene presence (at least in the provided subset results) is relatively sparse. It is therefore difficult to interpret and compare the data. Could you provide a pathway coverage information as numerical values? It would also be interesting to see pathways of the 20 samples from the bioreactor project as a figure.

6. Bioreactor example:

a. Did the re-assembly reduce N50 for some of the bins?

b. Did it change the quality?

c. Why is the re-assembly only performed for 5 samples?

Minor

• There is no quality control implanted for long reads (except length filtering). Doesn’t it make sense to remove reads that come from contaminants or have low quality?

• The Author summary paragraph is not the same as author contribution.

• Novelty: L71: It seems as if the authors make the indirect claim that muffin the first pipeline for hybrid assembly, is this true?

• The sentence L23-25 in the abstract is difficult to read. propose reformulation: “and can be further used for quantification and annotation by providing RNA-Seq data (optionally).”

• Is it possible to assemble with long reads only or does one need to define always short reads for the polishing?

• L98: Are Pac Bio long also supported?

• L127: Is it possible to update the GTDB ?

• L129: The sentence "GTDB substantially improved overall downstream results40" is misleading. One would assume that the sentence is about the downstream results of MUFFIN. But then the authors should explain which downstream results and how instead citing ref40.

• The authors look ahead on potential future version conflicts. However, I don't fully understand the solution they propose. How can a user make sure that they pick the correct version of nextflow or Muffin? Isn't there another way where it's not up to the user to read the release notes to exclude version conflicts e.g. a conda package providing the tested version of muffin and nextflow?

• It's confusing that the '-profile' command-line argument has only one leading hyphen which is inconsistent with the rest of the CLI.

Reviewer #3: The authors devised a particular metagenomic analysis workflow and applied it to 20 samples of public data. The workflow is composed of popular software tools and the orchestration software. The manuscript is focused on presenting this particular workflow, yet without giving any justification for the choices or discussion of alternatives. The description of the analysed samples is superficial to the point of being non informative.

Thus I struggled to apply criteria for publication in PLOS Computational Biology here:

• Originality - seems to be mostly related to hybrid assembly from short and long reads, which is commonly done when such data exists, which on itself is not very common.

• Innovation - all the components are public 3rd party tools, thus it seems to be mostly related to a particular configuration of the pipeline.

• High importance to researchers in the field - likely not.

• Significant biological and/or methodological insight - no.

• Rigorous methodology - alternatives exist for each step, yet not evaluated.

• Substantial evidence for its conclusions - no conclusions.

It reads as a part of documentation rather then a research article.

Apologies for not being able to be more positive.

**Have all data underlying the figures and results presented in the manuscript been provided?**

Reviewer #1: Yes

Reviewer #2: **No: **could you provide the results from the 20 sample bioreactor or at least the important part of it?

Reviewer #3: **No: **missed the data for Figure 3.

PLOS authors have the option to publish the peer review history of their article (what does this mean?). If published, this will include your full peer review and any attached files.

Reviewer #1: **Yes: **C. Titus Brown

Reviewer #2: **Yes: **Silas Kieser

Reviewer #3: No
---

## [Decision Letter · Decision Letter 1]

17 Jan 2021

Dear %TITLE% Van Damme,

We are pleased to inform you that your manuscript 'Metagenomics workflow for hybrid assembly, differential coverage binning, metatranscriptomics and pathway analysis (MUFFIN)' has been provisionally accepted for publication in PLOS Computational Biology.

Best regards,

Mihaela Pertea

Software Editor

PLOS Computational Biology

Mihaela Pertea

Software Editor

PLOS Computational Biology

Reviewer's Responses to Questions

**Comments to the Authors:**

Reviewer #2: The corrections are satisfying and I managed to run MUFFIN.

**Have all data underlying the figures and results presented in the manuscript been provided?**

Reviewer #2: Yes

PLOS authors have the option to publish the peer review history of their article (what does this mean?). If published, this will include your full peer review and any attached files.

Reviewer #2: **Yes: **Silas Kieser

---

## [Editor Report · Acceptance letter]

4 Feb 2021

PCOMPBIOL-D-20-01293R1 

Metagenomics workflow for hybrid assembly, differential coverage binning, metatranscriptomics and pathway analysis (MUFFIN)

Dear Dr Van Damme,

I am pleased to inform you that your manuscript has been formally accepted for publication in PLOS Computational Biology. Your manuscript is now with our production department and you will be notified of the publication date in due course.

With kind regards,

Alice Ellingham
